# Technological Solutions to Decrease Protein Content in Piglet Weaning Feed, Improving Performance, Gut Efficiency, and Environmental Sustainability

**DOI:** 10.3390/ani15121720

**Published:** 2025-06-11

**Authors:** Michele Lanzoni, Paul De Smet, Giovanni Ferrari, Davide Bochicchio, Sujen Santini, Valerio Faeti

**Affiliations:** 1DVM, Nutritionist and Farm Consultant, 47032 Bertinoro, Italy; 2Head Nutritionist Bivit Company, 8560 Wevelgem, Belgium; paul@bivit.com; 3Nutritionist, Neofarma Company, 47020 Longiano, Italy; giovanniferrari@neofarma.it; 4Research Centre for Animal Production and Aquaculture, Council for Agricultural Research and Economics, Via Beccastecca 345, 41018 Modena, Italy; davide.bochicchio@crea.gov.it (D.B.); sujen.santini@crea.gov.it (S.S.); valerio.faeti@crea.gov.it (V.F.)

**Keywords:** amino acids, weaning feed, low-protein diet, nitrogen efficiency

## Abstract

This study was conducted within the Nutritional Engineering in Weaning for Performance, Immunity, and Gut Efficiency (N.E.W. P.I.G.) The project is a plan aimed at developing innovative diets for weaning piglets that are characterized by low protein levels, achievable through the use of seven synthetic amino acids, while testing the elimination of fish meal and blood derivatives. The trial demonstrated that it is possible to improve the efficiency of the gut, avoiding the use of mass antibiotic treatments in weaning, achieving higher performance, reducing the inclusion of proteins in feed by valorizing by-products, and reducing the environmental impact linked to nitrogen excretion.

## 1. Introduction

Intestinal diseases are one of the most common problems in pig production, particularly in the immediate period following weaning, which is characterized by a higher risk of post-weaning diarrhea (PWD) [1]. The ban on the use of zinc and the need to gradually reduce the use of medicated premixes in feed in favor of more targeted treatments in water and by individual injection have imposed the adoption of new nutritional practices during the weaning phase. Among these, low-crude protein nutritional strategies appear to have a direct impact on the immune status of pigs [1], the potential to prevent or treat stress-related health problems [2], and on the reduction in PWD frequency [3,4,5,6,7,8,9,10,11,12,13,14,15]. The objective of precision nutrition is to prevent unwanted nutrients from arriving undigested in the most distal parts of the intestine, thereby removing them from the fermentations of pathogenic bacteria [16]. To reach this target, an accurate calculation of animal requirements in each production phase and raw materials nutrient matrix is needed. Since the undigested nutrients accumulate in excretions, the imbalance between dietary nutrient intake and needs also has a negative impact on the environment. In fact, the low-crude protein diet strategy is also an established strategy to reduce nitrogen emissions [17,18]. The key to more efficient feed utilization is first to have accurate requirements for each nutrient for every productive phase. Second, to use robust and accurate feed evaluation systems [19]. These criteria could be especially relevant in high crude protein diet formulation in the post-weaning period, where an excess of protein leads to a significant increase in the risk of digestive disorders and, consequently, alterations in the state of health [20]. Therefore, the low-raw protein diet represents one of the main strategies to reach the UN Sustainable Development Goals [21], including animal health and welfare, contrasting antibiotic resistance, and reducing the environmental impact of livestock production. The aim of the present study was to explore new nutritional engineering in weaning piglets to improve the health of the animals, increase nitrogen retention, and therefore feed efficiency. Moreover, it demonstrates that an entirely vegetable diet (excluding dairy by-products) is as effective as one containing fish meal and plasma, two cornerstones considered essential in this weaning phase. Our results suggest that a well-balanced vegetable diet with synthetic amino acids can outperform traditional high-protein feeds, offering health, performance, and environmental benefits.

## 2. Materials and Methods

### 2.1. Animals and Feeding Thesis

The trial involved 180 weaned piglets (Italian Duroc × Italian Large White) with an average age of 24 days, divided into 18 pens (3.3 m^2^ each) of 10 animals, distributed evenly among the three feeding treatments homogeneously based on litter, gender, live weight, and age. For each treatment, six pens were used, three of which were males and three of females. For each sex, there was one pen of heavy animals, one of medium, and one of light, to avoid excessive competition at the feeder and consequent loss of uniformity of the litter. Feeds were administered manually, and to avoid possible human mistakes in feed distribution, piglets were housed in three adjacent weaning rooms. The project compared three feeding treatments (Table 1 and Table 2), each with two post-weaning feeds, the first period 6–12 kg (32–45 days of age) and the second period 12–25 kg (45–69 days of age):-Control Feed (C): High-quality and low-protein commercial feeds studied for an antibiotic-free supply chain, containing plasma and fishmeal;-Thesis 1 (T1): Low-protein experimental feed with 7 synthetic amino acids, containing plasma and fishmeal;-Thesis 2 (T2): Low-protein experimental feed with 7 synthetic amino acids, without plasma and fishmeal.

The feeding was managed as follows: A single creep feed under the sow during lactation, starting at 10 days of age. One week before weaning, it was administered ad libitum, a common pre-starter feed until the start of the trial (32 days of age of piglets), during which the different feeds were administered ad libitum up to approximately 25 kg (69 days of age). The piglets were weighed individually after weaning to organize the groups in a homogeneous way (24 days of age). After seven days, they were reweighed for the start of the test with the first feed (32 days of age). After 13 days, they were reweighed for the change in feed (45 days of age), and finally, they were weighed at the end of the test period (69 days of age).

The trial began after a convenient period of adaptation to weaning conditions, so as not to overlap the stress of forming trial groups, the stress of weaning, and the stress of changing diet. In this way, at the beginning of the trial, all piglets were well adapted to the solid diet, with all stressful events left behind.

### 2.2. Data Collection and Calculation

The health of the piglets was monitored daily. A score was assigned to each cage/pen based on the number of cases and the extent (mild, medium, severe) of diarrheal phenomena in piglets, visually assessed from the consistency of the feces [13,22] (Table 3).

According to the G. Xiccato method [23], we evaluated nitrogen (N) efficiency during the production cycle considered. Calculating the difference between the final and initial weight of the piglets and comparing it to the amount of nitrogen ingested with the feed, we can obtain the percentage of fixed nitrogen, as the following equations:N efficiency (%) = {[ N animal final weight (*) − N animal initial weight (*)]/N intake (g)} × 100(1)

(*) represents the quantity of nitrogen present in the animals according to a fixed coefficient per weight category, which, for individuals weighing less than 40 kg, corresponds to 27 g of N per kg of live weight [21].

### 2.3. Statistical Analysis

All data were analyzed through the analysis of variance with the GLM procedure of SAS version 9.4 for Windows (SAS Institute Inc., Cary, NC, USA). The means value was estimated with the LSMEANS procedure of the SAS GLM. Variables that differed by *p* ≤ 0.05 were tested with the Bonferroni post hoc test, using alpha = 0.05 and alpha = 0.01, respectively.

Specifically, data distinguished by lowercase letters “a” and “b” indicate significance at *p* < 0.05, while data distinguished by uppercase letters “A” and “B” indicate significance at *p* < 0.01.

The live results were processed, taking repetition as the minimum cell (4 repetitions per thesis) with an analysis of variance according to the following model:Yijk = M + Ai +Bj + (AB)ij Eijk
whereYijk = dependent variable observed on the kma replication of the ijmo subgroup;M = overall mean;Ai = diet (i = 1.3);Bj = sex (j = 1.2);ABij = interaction thesis × sex;Eijk = residual error.
where the mean initial sex factor was considered as the blocking factor.

Biting episodes were analyzed using the Chi-Square and Fischer’s exact tests; the analysis was performed at the individual level (180 animals on the three tests and the presence or absence of tail lesions).

## 3. Results

### 3.1. Diarrhea

No significant differences were recorded in the diarrhea data. One mild case of diarrhea occurred in the T1 group during the first phase, and in the T2 during the second phase.

### 3.2. Tail Biting

In the first trial period, no cases of biting were recorded in any pen. In the second trial period, 51 cases of biting were recorded and distributed, as shown in Table 4. For both feeding phases, T2 is connected to the absence of biting with a probability lower than *p* < 0.01.

### 3.3. Average Weight

At the end of the first phase (32–45 days of age), a significant difference (*p* < 0.01) was noted between the two treatments (T1 and T2) and the control feed. Regarding the second phase (12–25 kg), the piglets belonging to the T2 group had a significantly higher weight than those that assumed the control feed, while the weights of the piglets receiving the T1 feed were comparable to those of T2 and the control (Table 4).

### 3.4. Feed Intake

Feed consumption reflects the trend of the animals’ weight gain but with a lower significance (*p* < 0.05). In the first phase, T1 and T2 treatments were significantly more consumed than the control feed, while in the second phase, only the T2 treatment was significantly more consumed than the control feed (Table 4).

### 3.5. Feed Conversion Rate (FCR)

FCR in the first period of the two treatments, T1 and T2, is significantly (*p* < 0.01) lower than that in the control treatment. In the second phase, the feed conversion index is lower in the control and T2 groups compared to the Thesis T1 (*p* < 0.05). Considering the entire test period (phase 1 and phase 2), the Thesis T2 appears to have a significantly lower feed conversion index (*p* < 0.05) compared to the control and T1 (Table 4).

### 3.6. Average Daily Gain (ADG)

ADG reflects the trend of the average weight of the treatments: in the first phase, the piglets of the T1 and T2 treatments showed an increase of 0.31 kg/day versus 0.23 kg/day of the control feed with a significance of *p* < 0.01. In the second phase, the group fed with Thesis 2 performed an ADG of 0.64 kg/day, significantly higher (*p* < 0.01) than that of the group fed with the control feed, while the group fed with Thesis 1 did not appear different from the other two groups. The same trend can be appreciated in the calculation of the ADG of the entire test period, where the T2 group was significantly higher (0.51 kg/day) (*p* < 0.01) than the control group (0.43 kg/day), while the group fed with the T1 thesis was not different from the other two groups (Table 4).

### 3.7. Nitrogen Efficiency

From the statistical processing, we observe that the T2 thesis fixed a significantly greater amount of nitrogen (*p* < 0.01) than the T1 thesis and the control group (T2 58.71% vs. T1 54.13% and C 54.02%) (Table 5).

## 4. Discussion

In a low-protein diet for weaning pigs, reconstituting the ideal protein using synthetic amino acids can obtain superior performance compared to a feed, even with a high formulation level characterized by a higher protein content [4]. The reason the difference in crude protein levels between the trial groups and control group is moderate derives from the fact that it was decided to compare two test feeds with a control feed that was already commercially established, inserted in antibiotic-free supply chains, already oriented towards zinc removal and mass treatments, and which had already demonstrated satisfactory performance. The aim of this paper is to assess which technical solution is more efficient in the weaning phase, pushing towards further protein reduction and removal of animal-origin raw materials (excluding dairy by-products). The three feeds are commercially available, and some features are covered by industrial confidentiality.

The control and T1 groups had in common the presence of fishmeal and plasma, raw materials absent in T2. The T2 feed, without raw materials of animal origin (excluding dairy by-products), obtained significantly higher results than the control feed for all the zootechnical parameters considered and in every weaning phase: average daily gain, feed intake, and feed conversion rate.

The T1 feed, containing fishmeal and plasma but formulated following the same principle as T2, followed the trend of T2 only in the first weaning phase (32–45 days of age). In the second phase (45–69 days of age), the T1 group did not maintain either the ingestion or growth rates of T2. This compromised the results, yielding substantially intermediate outcomes between the control and T2 groups. Since the T1 feed is formulated with synthetic amino acids and has the same balance as T2, the results in the second phase may have been influenced by the presence of fish meal, but more likely by the Deoxynivalenol levels recorded in this feed during this phase and the tail-biting episodes that occurred in the second weaning period. The results regarding tail biting in fact are, in fact, very consistent, and the T2 nutritional solution recorded a statistically significant result (*p* < 0.01). As observed by Minussi I. et al. [24], in low-protein diets, supplementation with adequate quantities of synthetic amino acids can limit the phenomenon of tail biting. This is certainly true for T2, but the same should also be true for T1. An unknown environmental component may have influenced T1 as a source of stress. The results obtained with the T2 formulation demonstrate that the presence of plasma and fish meal is not an essential element in the diet of weaning piglets. In fact, compared to a vegetable feed correctly balanced in its amino acid profile, the T1 formulation yielded worse results. The calculation of nitrogen retention demonstrates that, in the balance between nitrogen fixed in the muscle protein and nitrogen introduced with the feed, the vegetable diet characterized by a low level of protein and the use of seven synthetic amino acids is the most efficient and environmentally respectful solution.

## 5. Conclusions

The need to reduce the use of antibiotics and eliminate zinc at pharmacological levels requires the adoption of lower-protein diets during weaning [1]. This approach also aligns with the need at the zootechnical level to limit nitrogen emissions into the environment.

These objectives can be achieved through the correct reconstitution of the ideal protein by balancing the supply of synthetic amino acids. A correct and balanced low-protein formulation, respecting the needs of the animals from the perspective of precision nutrition, allows for the production of more efficient feed even than diets with a higher protein content and containing raw materials of animal origin, such as plasma and fishmeal. In addition to better zootechnical results and lower nitrogen emissions into the environment, piglets show reduced levels of stress and fewer aggressive behaviors, such as tail biting.

## Figures and Tables

**Table 1 animals-15-01720-t001:** The first period (32–45 days of age) feeds composition and analysis.

	Control	T1	T2
Barley (%)	30	35	35
Biscuit meal (%)	10.2	5	7.5
Soft wheat (%)	8		
Wheat flakes (%)	7.4		
Corn (%)	3	9	7
Corn flakes (%)		15	15
Wheat bran (%)	5	5	2.8
Dehulled oat (%)	5	5	7.5
Fishmeal (%)	3.5	3.5	
Concentrated soybean meal (60% raw protein) (%)	3.2		4
Plasma (%)	1	1	
High digestible protein sources ^1^ (%)	5.5	7.4	8.6
Milk by-products (%)	8	4.25	4.6
Beet pulp (%)		2.5	1
Sugars (saccarose, dextrose) (%)	1	0.5	0.5
Coconut oil (%)	0.5	0.6	0.65
Vitamins & trace-minerals (%)	0.25	0.25	0.25
Inorganic and organic calcium sources (%)	4.25	1.25	0.75
Salt (%)	0.45	0.45	0.45
MCP (%)	0.8	0.55	0.6
Organic acids (%)	1.1	1.2	1.2
Full herb, dried (%)	0.5		
Aminoacids (%)	0.67	2.3	2.3
Probiotics ^2^ (%)	0.3	0.3	0.3
Other additives (%)	0.33		
Crude protein (%)	16.33	15.62	15.45
Fats and oils (%)	4.86	3.92	3.88
Cellulose (%)	3.3	4	3.68
Starch (%)	39.1	43.24	41.34
Ash (%)	5.01	4.22	4.66
Biotin (mg)	0.10	0.3	0.3
Vit B1 (mg)	3.29	3	3
Vit C (mg)	40	105	105
Folic acid (mg)	1	1.5	1.5
Vit B6 (mg)	2.97	6	6
Vit A	15,000	15,000	15,000
Vit D3	2000	1995	1995
Calcium D-pantothenate (mg)	16.44	25	25
Vit B2 (mg)	6.97	9.88	9.88
Vit B12 (mg)	0.04	0.05	0.05
Niacinamide (mg)	32.46	35	35
Vit E (mg)	100	101	101
Vit K3 (mg)	2.36	10	10
Betaine (mg)	156	427.5	427.5
Cu (mg)	90	150	150
Fe (mg)	170	97	97
Mn (mg)	50	41.7	41.7
I (mg)	1.5	1.9	1.9
Zn (mg)	105	72	72
Se (mg)	0.42	0.42	0.42
Aflatoxins (ppm)	1.93	0.7	1.53
Deoxynivalenol (ppm)	0.59	0.57	0.68

^1^ High-digestible protein sources in control: potato protein, full-fat extruded soybean meal; in T1 e T2: whey protein concentrate, potato protein, full-fat extruded soybean meal, dried yeasts. ^2^ Probiotic in control: Enterococcus Faecium 299,900,000 CFU; Probiotic in T1: Saccharomyces Cerevisiae 1,000,000,000 CFU; Probiotic in T2: Saccharomyces Cerevisiae 1,782,000,000 CFU.

**Table 2 animals-15-01720-t002:** The second period (45–69 days of age) feeds composition and analysis.

	Control	T1	T2
Barley (%)	30	30	30
Biscuit meal (%)	10		10
Soft wheat (%)	6.4		
Wheat flakes (%)	4		
Corn (%)	18	43.6	30
Wheat bran (%)	7.7	2.5	5.5
Dehulled oat (%)	2		
Fishmeal (%)	5	4	
Concentrated soybean meal (60% raw protein) (%)	1.5		4.4
Soybean meal 47% (%)	3	9	9
High digestible protein sources ^1^ (%)	3	3.4	3.4
Milk by-products (%)	2		
Beet pulp (%)		2.5	2.5
Sugars (saccarose, dextrose) (%)		0.4	0.4
Soy oil (%)	0.7	1	1
Coconut oil (%)	0.3		
Vitamins and trace minerals (%)	0.25	0.25	0.25
Inorganic and organic calcium sources (%)	3.05	1.15	1.25
Salt (%)	0.5	0.4	0.4
Monocalcium Phosphate (%)	0.5	0.5	0.6
Organic acids (%)	0.6		
Full herb, dried (%)	0.6		
Aminoacids (%)	0.9	1.3	1.3
Probiotics ^2^ (%)	0.3	0.3	0.3
Crude protein (%)	16.05	15.51	15.85
Fats and oils (%)	4.71	4.25	4.05
Cellulose (%)	3.69	4.28	4.77
Starch (%)	41.08	45.66	41.65
Ash (%)	4.85	4.28	4.54
Biotin (mg)	0.05	0.3	0.3
Vit B1 (mg)	1.5	3	3
Vit C (mg)	26.25	105	105
Folic acid (mg)	0.69	1.5	1.5
Vit B6 (mg)	2	6	6
Vit A	13,500	15,000	15,000
Vit D3	2000	1995	1995
Calcium D-pantothenate (mg)	13.5	25	25
Vit B2 (mg)	5.5	9.88	9.88
Vit B12 (mg)	0.03	0.05	0.05
Niacinamide (mg)	27.5	35	35
Vit E (mg)	77.5	101	101
Vit K3 (mg)	1.35	10	10
Betaine (mg)	125	427.5	427.5
Cu (mg)	90	150	150
Fe (mg)	165	97	97
Mn (mg)	50	41.7	41.7
I (mg)	11.5	1.9	1.9
Zn (mg)	102.5	72	72
Se (mg)	0.42	0.42	0.42
Aflatoxins (ppm)	0.24	0.59	/
Deoxynivalenol (ppm)	not detected	0.8	not detected

^1^ High-digestible protein sources in control: potato protein, full-fat extruded soybean meal, in T1 e T2: full-fat extruded soybean meal. ^2^ Probiotic in control: Saccharomyces Cerevisiae 260,000,000 CFU; Probiotic in T1: Saccharomyces Cerevisiae 1,000,000,000 CFU; Probiotic in T2: Saccharomyces Cerevisiae 1,782,000,000 CFU.

**Table 3 animals-15-01720-t003:** Diarrhea scoring methodology: scores for diarrhea incidence assigned to each cage/pen on the basis of the fraction of litter suffering from diarrhea and diarrhea intensity.

Fraction of Litter ^1^ Suffering from Diarrhea	Diarrhea Intensity		
	Mild	Medium	Serious
0	0	0	0
1/3	1	4	7
2/3	2	5	8
3/3	3	6	9

^1^ 10 piglets per cage/pen. Piglets suffering from diarrhea were treated parenterally with the antibiotic enrofloxacin. All cases of tail and ear biting, such as all individual and group therapeutic treatments during the feed trial period were recorded and data were collected in the stable register. Piglets suffering from tail and ear biting were treated parenterally with the antibiotic rifaximin.

**Table 4 animals-15-01720-t004:** Tail-biting data.

	Pen1	Pen2	Pen3	Pen4	Pen5	Pen6	Tot Tail Bites
Control	6	10	0	1	0	10	27 ^A^
T1	2	10	3	0	0	9	24 ^A^
T2	0	0	0	0	0	0	0 ^B^

AB Different letters within a row indicate significant differences (*p* < 0.01).

**Table 5 animals-15-01720-t005:** Weight, Average Daily Gain, Feed Intake, FCR, Nitrogen efficiency.

	Control	T1	T2
Average Weight at 32 days (Kg)	8.40	8.46	8.47
Average Weight at 45 days (Kg)	11.90 ^B^	13.09 ^A^	13.08 ^A^
Average Weight at 69 days (Kg)	24.57 ^B^	26.05 ^AB^	27.82 ^B^
Average Daily Gain 32–45 days (Kg/d)	0.23 ^B^	0.31 ^A^	0.31 ^A^
Average Daily Gain 45–69 days (Kg/d)	0.55 ^B^	0.56 ^AB^	0.64 ^A^
Average Daily Gain 32–69 days (Kg/d)	0.43 ^B^	0.46 ^AB^	0.51 ^A^
32–45 Feed Intake (Kg)	7.43 ^b^	7.97 ^a^	7.91 ^a^
45–69 Feed intake (Kg)	23.18 ^b^	25.26 ^ab^	25.79 ^a^
Feed Conversion Rate 32–45 days	2.12 ^A^	1.72 ^B^	1.72 ^B^
Feed Conversion Rate 45–69 days	1.83 ^b^	1.95 ^a^	1.75 ^b^
Feed Conversion Rate 32–69 days	1.89 ^a^	1.89 ^a^	1.74 ^b^
Nitrogen efficiency (%)	54.02 ^B^	54.13 ^B^	58.71 ^A^

AB Different letters within a row indicate significant differences (*p* < 0.01). ab Different letters within a row indicate significant differences (*p* < 0.05).

## Data Availability

The original contributions presented in this study are included in the article. Further inquiries can be directed to the corresponding author.

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
