# Peer review of "Technological Solutions to Decrease Protein Content in Piglet Weaning Feed, Improving Performance, Gut Efficiency, and Environmental Sustainability"

_animals, 2025, doi:10.3390/ani15121720_

Round 1

Reviewer 1 Report

Comments and Suggestions for Authors

Summary:

Authors are evaluating low-protein diets, specifically removing blood and fish meals from the diets, and assessing the growth of the piglet. Three diets were assessed: a control, low-protein with blood and fish meal, and low-protein without blood and fish meal. The impact would be that the piglets have better growth performance post weaning and there is environmental benefits from a decrease in nitrogen excretion. It was found that a low protein diet without fish and blood meal has higher average daily gain, improved feed conversion ratio, and nitrogen utilization. Overall, demonstrating that a low-protein diet can improve growth performance post-weaning and have a positive environmental impact.

Comments:

Title: A small recommendation to adjust the title of your manuscript, as you did not analyze gut health in this project.

Abstract

Lines 26-29 Results were  “significantly better.” Please be specific about what better means to you. Is this higher?

Abstract general comment: Do you have space to include an overall conclusion? If not, it would not, no issue, but I think it could be beneficial to the quality of the abstract.

Introduction

Sentence on lines 38 – 40 Citation needed

Sentence on line 44-48 Please re-write this sentence, there are some grammatical issues, particularly in the first half of the sentence. Apart from that, this principle that is being described strictly for low protein diets, is something that is considered in all diet formulations. Nutritionists strive to have accurate production phase diets that are precise, in addition to diets that contain only what the animal needs. I believe the point that you are trying to make is something along the lines of feeding amino acids can lead to an even more precise diet, which I would agree with. In all, I think I see the point you are trying to make, but I would argue that this sentence needs to be re-written.

Lines 51-56: There is nothing wrong with the point you are trying to make with this sentence. But, grammatically, it is a very long sentence that could be easily broken up into two or three separate sentences that would be easier to read.

Lines 61-62: How does improving feed efficiency also lead to maintaining performance? If you are improving feed efficiency, you are increasing performance.

Lines 63-64: In the abstract, you talk of fish meal. Here you are specifying herring, I would recommend using fish meal so that it is more applicable to all geographical locations.

Line 65: Be consistent on the terms that are being used. You just called this diet “vegetal feed” and now you are calling it “vegetable diet.” I would use the same terminology for the whole paper.

M&M

Line 71: Abstract says the average age is 27  days, here is says 24 days

Line 71: What were the dimensions of the pen?

Lines 71-73:  It is mentioned, but could you make it clearer that the pens are same-sex? This is a very small thing, but I think it might help.

Line 75: I believe you mean nonuniformity, and not deformity.

Line 76-77: The piglets began study at age 24 , but phase 1 does begin until day 32? What occurred between day 24 until day 32?

Lines 84-84: You might need to explain more what a “single feed under the sow” means, this might be geographical differences, but is this the same as putting the feed in the farrowing crate with the piglets while they are still nursing? If so, please further explain if the feed is constantly available while still with the sow, or if it is a limited amount available.

To build on this, until now, it was believed that the whole study is postweaning, but if the piglets are fed feed while still suckling, then this study actually begins pre-weaning. Could you clarify this?

Lines 87-88 What did the pigs eat for the first week after weaning? Also, from day 24 to day 32, is not six days, I think you could get away with calling it seven, but it is technically eight, based on how you calculated days between day 32 and 45.

Tables 1 and 2. Please add footnotes to describe the following: High digestible protein sources, what is the source?; premix, what is in the premix and what are the inclusions of each ingredient?; calcium sources, what sources?; MCP, spell out the abbreviation; organic acids, which organic acids?; natural compounds, what compounds and how much of each?; probiotics, what is the source? Which strains? How much of each strain?; other additives, what are these additives and how much of each?

Line 99: I think you have left a note to yourself in the manuscript, but I agree you should add the method 😊

Line 104- 105 and Lines 113-115: citation did not transfer in correctly, please fix.

Line 132: Small typo here, “(j=1.2);” I believe the period should be a comma

General comment: Wonderful statistic section, very nicely written, great job.

Results

Line 144: By box do you mean cage?

Line 146: “fases” should be “phases”

Table 3. Again, should box be cage? In the materials and methods, the pigs were in cages; otherwise, please explain what a box is. Additionally, what happened to box 4 and 5? I would recommend using superscript to denote the statistical differences, and you will have to add the statistical meaning again as a footnote. Remember, all tables and figures should contain enough information to stand alone outside of a manuscript.

General comment for the whole results section: You refer to your theses as treatments in the results, but up until this point they have been called theses. I am fine with either name, but please pick one and stick with it throughout the manuscript. I will not point out every spot you use treatment, but please go through and fix this.

General comment: Please capitalize the word table when referring to a Table in the manuscript. Again, please go through and fix this throughout the results section.

Table 4: I would recommend putting your statistical differences in superscript. Make sure to include a footnote that explains the abbreviations “T1”, “T2”, “ADG” and “FCR.” Again, all tables must be able to stand alone.

Discussion

Line 187: The diets were not higher in protein. In fact, the protein content in the thesis diet had about the same amount of protein as the control, according to the analysis you reported in Table 1.

Line 206-207 Were the pigs not exposed to the same environmental stressors? If they experienced different environments, this is a major confounding variable that needs to be addressed. Were all the pigs housed in the same room at the same time? If not, this needs to be stated in the materials and methods.

Line 208-209 I do not think this sentence is reading like you anticipate, to me, it is stating that a vegetable only diet is worse than one with fish and blood meal. Please re-write this sentence as you just proved that this is not true.

Conclusions

Lines 216-218: While you measure diarrhea, overall health of the piglets was not fully assessed here in. Making a conclusion that your study showed a way to reduce antibiotic use I think is not appropriate here as health of the piglets was not fully analyzed. I.e. intestinal morphology, intestinal permeability, fecal analysis for disease shedding.

In fact, the only report of diarrhea in the study was in the low-protein diets, which would make your previous statement false.

Line 223-225: This is an incomplete sentence.

Major Concern: One theme that is brought up in the introduction and conclusion is the improvement of health post-weaning. But, the majority of diarrhea that occurs post-weaning is in the first week post-weaning. In this study, that week, from when the piglets are removed from the sow and into the nursery,  does not appear to be evaluated for these piglets. This is a major oversight in the study and needs to be addressed. This week of missing data is critical for the work that is being presented here. Major justification for this is needed.

Furthermore, this study is evaluating the effects of diets, yet in your diet table you have omitted critical information to the composition of your diets. For example, you listed “probiotics,” which is not something that is included in all nursery diets. There are also major discrepancies between the inclusion of ingredients across diets. For example, only the control diet contains “natural compounds” and “other additives”. As a scientist, I want to know 1) what these things are and 2) why are they not in T1 and T2? In short, this just adds more confounding variables. I would expect to only see differences across diets in 1) the fish meal, 2) the plasma, and 3) a select few ingredients (i.e., soy and barley). These diets are vastly different from each other.

Overall: I think this is an interesting paper conceptually, and it makes very good points, but major work needs to be done, and justifications for major differences in diets and the missing data from the first week post-weaning.

Author Response

Reviewer 1

Comments:

Title: A small recommendation to adjust the title of your manuscript, as you did not analyze gut health in this project.

Proposed new title: Technological solutions to decrease protein content in piglet weaning feed improving performance, gut efficiency, and environmental sustainability.

Abstract

Lines 26-29 Results were  “significantly better.” Please be specific about what better means to you. Is this higher? .

“Significantly higher”

Abstract general comment: Do you have space to include an overall conclusion? If not, it would not, no issue, but I think it could be beneficial to the quality of the abstract.

Abstract changed in

“The trial explored innovative low protein diets for weaning piglets, testing the elimination of fish meal and blood derivatives. The trial compared 3 treatments, each with two post-weaning feeds. Control feed with blood plasma and fishmeal, T1: low-protein feed with 7 aminoacids with fishmeal and blood plasma, T2: low-protein feed with 7 aminoacids without fishmeal and blood plasma. The trial tested 180 weaned piglets at 24 days of age evenly distributed on weight, age and litter of origin. At the end of the first phase, T1 and T2 recorded a significantly higher average weight, ADG and FCR compared with the control feed. At the end of the second phase, T2 group maintained a significantly better weight, ADG and FCR than control group. Considering the entire test period (phase 1 and phase 2) the T2 thesis appears to have a significantly higher average weight, ADG and lower FCR compared to the Control group. T2 thesis fixed a significantly (P<0,01) greater quantity of nitrogen compared to the T1 thesis and Control feed (T2 58.71% vs T1 54.13% and Control 54.02%). In conclusion a low-protein diet without raw materials of animal origin proved more efficient in terms of performance and nitrogen retention”

Introduction

Sentence on lines 38 – 40 Citation needed done

Sentence on line 44-48 Please re-write this sentence, there are some grammatical issues, particularly in the first half of the sentence. Apart from that, this principle that is being described strictly for low protein diets, is something that is considered in all diet formulations. Nutritionists strive to have accurate production phase diets that are precise, in addition to diets that contain only what the animal needs. I believe the point that you are trying to make is something along the lines of feeding amino acids can lead to an even more precise diet, which I would agree with. In all, I think I see the point you are trying to make, but I would argue that this sentence needs to be re-written.

Corrected in “The objective of precision nutrition is to avoid unwanted nutrients arrive undigested in the most distal parts of the intestine, removing them from the fermentations of pathogenic bacteria [15]. To reach this target, an accurate calculation of animal requirement in each production phase and raw materials nutrient matrix are needed. Since the undigested nutrients accumulate in excretions, the imbalance between dietary nutrient intake and needs also has a negative impact on the environment”.

Lines 51-56: There is nothing wrong with the point you are trying to make with this sentence. But, grammatically, it is a very long sentence that could be easily broken up into two or three separate sentences that would be easier to read.

Corrected in “The key for more efficient feed utilisation is to have first accurate requirements for each nutrient for every productive phase. Second, to use robust and accurate feed evaluation systems [18]. These criteria could be especially relevant in high crude protein diets formulation in post-weaning period. where an excess of protein leads to a significantly increase of the risk of digestive disorders and therefore alterations in the state of health [19]”

Lines 61-62: How does improving feed efficiency also lead to maintaining performance? If you are improving feed efficiency, you are increasing performance.

Corrected in “The aim of present study was to explore new nutritional engineering in weaning piglets to improve the health of the animals, increase nitrogen retention and therefore feed efficiency, resulting possibly in higher performance”.

Lines 63-64: In the abstract, you talk of fish meal. Here you are specifying herring, I would recommend using fish meal so that it is more applicable to all geographical locations.

Yes, fish meal instead of herring meal

Line 65: Be consistent on the terms that are being used. You just called this diet “vegetal feed” and now you are calling it “vegetable diet.” I would use the same terminology for the whole paper.

In all article it will be used the only term “vegetal diet”

M&M

Line 71: Abstract says the average age is 27  days, here is says 24 days

Correct, it’s 24 days.

Line 71: What were the dimensions of the pen? (3.3 m2 each)

Lines 71-73:  It is mentioned, but could you make it clearer that the pens are same-sex? This is a very small thing, but I think it might help.

we believe we have specified it sufficiently

Line 75: I believe you mean nonuniformity, and not deformity.

Corrected in “and consequent loss of uniformity of the litter.

Line 76-77: The piglets began study at age 24 , but phase 1 does begin until day 32? What occurred between day 24 until day 32?

On day 24 the groups were formed, on day 32 the trial started. It is specified!

Lines 84-84: You might need to explain more what a “single feed under the sow” means, this might be geographical differences, but is this the same as putting the feed in the farrowing crate with the piglets while they are still nursing? If so, please further explain if the feed is constantly available while still with the sow, or if it is a limited amount available.

To build on this, until now, it was believed that the whole study is postweaning, but if the piglets are fed feed while still suckling, then this study actually begins pre-weaning. Could you clarify this?

Added: “The trial began after a convenient period of adaptation to weaning conditions, so as not to overlap the stress of forming trial groups, stress of weaning and the stress of changing diet. In this way, at the beginning of the trial all piglets were well adapted to the solid diet and with all stressing events left behind”.

Lines 87-88 What did the pigs eat for the first week after weaning? Also, from day 24 to day 32, is not six days, I think you could get away with calling it seven, but it is technically eight, based on how you calculated days between day 32 and 45.

before the start of the trial they ate T2. We corrected 6 days with 7

Tables 1 and 2. Please add footnotes to describe the following: o rganic acids, which organic acids?; natural compounds, what compounds and how much of each?; probiotics, what is the source? Which strains? How much of each strain?; other additives, what are these additives and how much of each?

we have specified better “High-digestible protein sources”. It is not possible to provide more details as they are covered by the intellectual property of the company

Line 99: I think you have left a note to yourself in the manuscript, but I agree you should add the method ? Added

Line 104- 105 and Lines 113-115: citation did not transfer in correctly, please fix.

done

Line 132: Small typo here, “(j=1.2);” I believe the period should be a comma

done

General comment: Wonderful statistic section, very nicely written, great job.

Results

Line 144: By box do you mean cage?

Corrected box and cage in all paper in “pen”

Line 146: “fases” should be “phases”

done

Table 3. Again, should box be cage? In the materials and methods, the pigs were in cages; otherwise, please explain what a box is. Additionally, what happened to box 4 and 5? I would recommend using superscript to denote the statistical differences, and you will have to add the statistical meaning again as a footnote. Remember, all tables and figures should contain enough information to stand alone outside of a manuscript.

Corrected box and cage in all paper in “pen”

Re-numbered pens to avoid confusion

General comment for the whole results section: You refer to your theses as treatments in the results, but up until this point they have been called theses. I am fine with either name, but please pick one and stick with it throughout the manuscript. I will not point out every spot you use treatment, but please go through and fix this.

Corrected in all paper with “treatments”

General comment: Please capitalize the word table when referring to a Table in the manuscript. Again, please go through and fix this throughout the results section.

done

Table 4: I would recommend putting your statistical differences in superscript. Make sure to include a footnote that explains the abbreviations “T1”, “T2”, “ADG” and “FCR.” Again, all tables must be able to stand alone.

Done

Discussion

Line 187: The diets were not higher in protein. In fact, the protein content in the thesis diet had about the same amount of protein as the control, according to the analysis you reported in Table 1.

it is a bibliographic citation as better specified

Line 206-207 Were the pigs not exposed to the same environmental stressors? If they experienced different environments, this is a major confounding variable that needs to be addressed. Were all the pigs housed in the same room at the same time? If not, this needs to be stated in the materials and methods.

In Materials and methods added the sentence “Feeds were administered manually and to avoid possible human mistakes in feed distribution, piglets were housed in three adjacent weaning rooms.”

Line 208-209 I do not think this sentence is reading like you anticipate, to me, it is stating that a vegetable only diet is worse than one with fish and blood meal. Please re-write this sentence as you just proved that this is not true.

rewritten sentence

Conclusions

Lines 216-218: While you measure diarrhea, overall health of the piglets was not fully assessed here in. Making a conclusion that your study showed a way to reduce antibiotic use I think is not appropriate here as health of the piglets was not fully analyzed. I.e. intestinal morphology, intestinal permeability, fecal analysis for disease shedding.

In fact, the only report of diarrhea in the study was in the low-protein diets, which would make your previous statement false.

the association between low protein diets and intestinal health is an element confirmed by the literature, as better specified

Line 223-225: This is an incomplete sentence.

the reviewer's suggestion is not clear

Major Concern: One theme that is brought up in the introduction and conclusion is the improvement of health post-weaning. But, the majority of diarrhea that occurs post-weaning is in the first week post-weaning. In this study, that week, from when the piglets are removed from the sow and into the nursery,  does not appear to be evaluated for these piglets. This is a major oversight in the study and needs to be addressed. This week of missing data is critical for the work that is being presented here. Major justification for this is needed.

Furthermore, this study is evaluating the effects of diets, yet in your diet table you have omitted critical information to the composition of your diets. For example, you listed “probiotics,” which is not something that is included in all nursery diets. There are also major discrepancies between the inclusion of ingredients across diets. For example, only the control diet contains “natural compounds” and “other additives”. As a scientist, I want to know 1) what these things are and 2) why are they not in T1 and T2? In short, this just adds more confounding variables. I would expect to only see differences across diets in 1) the fish meal, 2) the plasma, and 3) a select few ingredients (i.e., soy and barley). These diets are vastly different from each other

In the discussion chapter added the sentence:

 “The reason why the difference in crude protein levels between trial groups and control group is moderate, derives from the fact that it was decided to compare two test feeds with a control feed that was already commercially established, inserted in antibiotic-free supply chains, already oriented towards zinc removal and mass treatments and which had already demonstrated satisfying performance. The aim of this paper is to assess which technical solution is more efficient in weaning phase pushing towards a further protein reduction and removal of animal-origin raw materials (excluding dairy by-products). The three feeds are commercially available, and some features are covered by industrial confidentiality"

Furthermore, where possible, some raw materials were better specified.

Overall: I think this is an interesting paper conceptually, and it makes very good points, but major work needs to be done, and justifications for major differences in diets and the missing data from the first week post-weaning

Reviewer 2 Report

Comments and Suggestions for Authors

Main comments:

  1. Please pay attention and follow the journal’s style throughout the manuscript including references, table headings, reference formatting, units etc.
  2. Clarify whether antibiotics or zinc oxide were used during the trial, as these could confound gut health outcomes. The abstract mentions "reducing antibiotic use," but the Methods section omits this detail.
  3. The trial was divided into two phases, each using distinct feeds. It is necessary to confirm whether the timing of phase transitions is reasonable and whether there are potential confounding factors. Additionally, with a sample size of 180 piglets divided into 18 pens (10 piglets per pen), was the variability among pens and the hierarchical structure of statistical analysis appropriately accounted for?
  4. The paper mentions documenting diarrhea and tail-biting incidents, but the scoring methodology for diarrhea is not described in detail.
  5. In Section 3.4, FCR results for T1 and T2 during Phase 2 are described as having "limited significance (P<0.05)," yet Table 4 labels them with lowercase letters (P<0.05).
  6. The claim that T1’s inferior performance in Phase 2 is linked to Deoxynivalenol levels lacks supporting data. Provide mycotoxin analysis results for all feeds to validate this hypothesis.
  7. While T2’s absence of tail biting is notable, the discussion should address potential confounding factors (e.g., environmental enrichment, stocking density) that might influence this outcome.
  8. Please check and modify the format of some references in the article.

Author Response

Reviewer 2

Main comments:

1. Please pay attention and follow the journal’s style throughout the manuscript including references, table headings, reference formatting, units etc.

we have corrected the inaccuracies

2. Clarify whether antibiotics or zinc oxide were used during the trial, as these could confound gut health outcomes. The abstract mentions "reducing antibiotic use," but the Methods section omits this detail.

During the experiment, no zinc oxide was used and the dierreas were treated according to the veterinarian's prescription. As suggested by the other reviewer, the concept of gut health has been removed. The mention of the reduction of antibiotic use refers to the specifically reported bibliography.

3. The trial was divided into two phases, each using distinct feeds. It is necessary to confirm whether the timing of phase transitions is reasonable and whether there are potential confounding factors.The trial is in a single phase and the change is consistent with the evolution of nutritional needs. Additionally, with a sample size of 180 piglets divided into 18 pens (10 piglets per pen), was the variability among pens and the hierarchical structure of statistical analysis appropriately accounted for? Yes, the statistical analysis was conducted as described starting from the "general model".

4. The paper mentions documenting diarrhea and tail-biting incidents, but the scoring methodology for diarrhea is not described in detail. Added

5. In Section 3.4, FCR results for T1 and T2 during Phase 2 are described as having "limited significance (P<0.05)," yet Table 4 labels them with lowercase letters (P<0.05).

fixed

6. The claim that T1’s inferior performance in Phase 2 is linked to Deoxynivalenol levels lacks supporting data. Provide mycotoxin analysis results for all feeds to validate this hypothesis.

where there was no value we better specified that "it was not detected". Ours remains a possible hypothesis as stated in the sentence

7. While T2’s absence of tail biting is notable, the discussion should address potential confounding factors (e.g., environmental enrichment, stocking density) that might influence this outcome.

the test was conducted excluding “potential confounding factors”

8. Please check and modify the format of some references in the article.

we have corrected the inaccuracies

Round 2

Reviewer 1 Report

Comments and Suggestions for Authors

I appreciate all the time and effort you've put into the revisions. I believe the manuscript is coming along nicely.

I still have some issues with the diets, while I understand the IP issue with the diets for including sources. From an animal science perspective, I still believe you should be able to include the exact amount of the inclusion of ingredients in the vitamin/mineral premix (i.e. ppm of vitamin in the mix, mg of mineral in the mix, etc). This is an important piece of information as vitamins and minerals play a large part of animal health. Additionally, which probiotic strains were given to the pigs. As diarrhea is a measure of the study, probiotics can have a major impact on this.

Additionally, I have two quick comments on the diarrhea information provided. However, Table 3 is not broken out by diet/treatment or phase. This leaves the reader without the chance to draw their own conclusions about the cases of diarrhea across diets/treatments. Secondly, there are several instances of diarrhea reported, including 9 cases of serious diarrhea for the entire litter. But it is stated in the results section there was "One mild case of diarrhea" for both T1 and T2. Could you further explain this?

Author Response

Reviewer 1 (round2)

I appreciate all the time and effort you've put into the revisions. I believe the manuscript is coming along nicely.

Comment 1: I still have some issues with the diets, while I understand the IP issue with the diets for including sources. From an animal science perspective, I still believe you should be able to include the exact amount of the inclusion of ingredients in the vitamin/mineral premix (i.e. ppm of vitamin in the mix, mg of mineral in the mix, etc). This is an important piece of information as vitamins and minerals play a large part of animal health. Additionally, which probiotic strains were given to the pigs. As diarrhea is a measure of the study, probiotics can have a major impact on this.

Answer 1: Added in tables 1 and 2.

Comment 2: Additionally, I have two quick comments on the diarrhea information provided. However, Table 3 is not broken out by diet/treatment or phase. This leaves the reader without the chance to draw their own conclusions about the cases of diarrhea across diets/treatments. Secondly, there are several instances of diarrhea reported, including 9 cases of serious diarrhea for the entire litter. But it is stated in the results section there was "One mild case of diarrhea" for both T1 and T2. Could you further explain this?

Answer 2: Table 3 is in the materials and methods section and indicates the scoring criterion for recording any diarrhea. Therefore "9" is the maximum score that can be achieved if severe diarrhea occurs throughout all the cage. This condition did not occur during the test.

Reviewer 2 Report

Comments and Suggestions for Authors

no comments

Author Response

Thank you very much for reviewing the manuscript.